# Robust Algorithm Software for NACA 4-Digit Airfoil Shape Optimization Using the Adjoint Method

**Naser Tanabi** , **Agesinaldo Matos Silva, Jr.** , **Marcosiris Amorim Oliveira Pessoa** and **Marcos Sales Guerra Tsuzuki** *

Computational Geometry Laboratory, Departamento de Engenharia Mecatrônica e de Sistemas Mecânicos, Escola Politécnica da Universidade de São Paulo, São Paulo CEP 05508-030, Brazil
* Correspondence: mtsuzuki@usp.br; Tel.: +55-11-3091-5759

**Abstract:** Optimizing the aerodynamic shape of an airfoil is a critical concern in the aviation industry. The introduction of flexible airfoils has allowed the shape of the airfoil to vary, depending on the flight conditions. Therefore, in this study, we propose an algorithm that is capable of robustly optimizing the shape of the airfoil based on variable parameters of the airfoil and flight conditions. The proposed algorithm can be understood as an optimization method, which employs the adjoint method, a powerful tool for estimating the sensitivity of the model output to the input in numerous studies. From an aerodynamic perspective, the development of shape geometry is a crucial step in airfoil development. The study used NACA-4 digit airfoils as input for the initial assumption and the range of shape change. The optimal shape was found using the proposed algorithm by defining one NACA profile as the initial value and another NACA profile as the limit for the optimized shape, considering the aerodynamic coefficients and flight conditions. However, morphing airfoils have certain deformation limitations. As an innovation in the algorithm, bounds were defined for the shape change during optimization so that the result can be constructed within the capabilities of the morphing wing. These bounds can be adjusted (depending on the capabilities of the airfoils). To validate the proposed algorithm, the study compared it with a previous flow solver for the same airfoil.

**Keywords:** gradient method; shape optimization; adjoint method

## 1. Introduction

Determining the geometric shape of an aircraft's wings is crucial for optimizing its aerodynamics and performance. Adjusting geometric constraints can improve various aspects, such as aerodynamic performance, flight distance, and flight time. The first steps in wing design involve defining the wing's geometry and selecting an appropriate airfoil type based on specific parameters. Special mission requirements may require designing and manufacturing the necessary parts and equipment to meet different operating conditions. The wing and airfoil are critical components, and numerical optimization techniques can be divided into two main categories: gradient-free and gradient-based methods. When dealing with large-scale aerodynamic optimization problems with multiple variables, the adjoint method is the most effective approach, particularly when the number of design variables exceeds 100 [1,2].

The focus of this paper is on robust aerodynamic shape optimization; we present an algorithm that differs from other similar articles. While other articles rely on commercial software or algorithms that do not consider flexible airfoil capabilities, this study takes into account the airfoil's global geometry instead of solely modifying the mesh points of its boundary without regard to its flexibility. Building upon previous research by Tanabi et al. [3], who utilized the pressure distribution around the airfoil to determine its shape [3], this work uses design parameters and flight conditions to determine the airfoil's shape, which is a more adaptable and versatile approach.

The primary objective of this study is to develop an algorithm that considers critical parameters, such as chord length, maximum thickness, and maximum camber, for flexible NACA-4-digit airfoils. The algorithm employs a unique reverse design approach to determine the shape of the NACA-4 digit airfoil based on the surrounding pressure distribution. During each iteration of the algorithm, it checks if the pressure distribution can generate NACA-4 digits while imposing constraints to ensure that the variables remain within acceptable ranges.

In this study, we conducted a comprehensive review of relevant literature and clarified the adjoint equation and optimization algorithm. The flow solver was validated by comparing its results to those obtained through numerical solutions. Finally, we present the results of the multipurpose optimization approach applied to an airfoil in a given regime, followed by a detailed discussion of the proposed algorithm's performance.

## 2. Bibliographic Review

The introduction of the gradient-based CFD method by Jameson [4] in 1988 marked the first application of adjoint equations to transient flow [4]. They are also among the most prominent developers of this method and have utilized adjoint equations in various computational fluid dynamics test cases, including the optimization of aerodynamic shapes for Eulerian flows that govern the Navier–Stokes equations. Lions was the first person to use optimization to solve differential equations [5], while adjoint equations were originally developed for shape optimization in fluid dynamics to solve a defined control problem [5,6]. The coupling of perturbation equations with this method was achieved by integrating the Reynolds averaged Navier–Stokes (RANS) equations with the Spalart–Allmaras single equation turbulence model [7,8]. Several researchers, including Giles et al. [9], utilized the adjoint optimization method in fluid dynamics and Eulerian equations and conducted significant research in this field [9,10]. Later, Giles and Pierce [11] used Green's functions for the adjoint equations [11]. To design an airfoil, they used the adjoint method under incompressible non-viscous conditions in the far-field region [12], while Xie [13] used the adjoint equations by considering the angle of attack based on the boundary conditions of the distant region [13].

In the multi-objective optimization field, various types of research studies have been conducted. Braibant and Fleury [14] presented a method for shape optimization utilizing B-splines, which are piecewise polynomial curves that can represent intricate shapes, making them valuable in engineering designs [14]. Poles et al. [15] investigated the effect of initial population sampling on the convergence of multi-objective genetic algorithms (MOGAs) [15]. The authors conducted experiments using different techniques for initial population sampling and evaluated their impact on the performance of MOGA. Additionally, they introduced a new evolutionary algorithm for multi-objective optimization, MOGA-II, and tested its performance on single-objective optimization problems, comparing it to two successful algorithms: differential evolution and a standard evolutionary algorithm. Mariotti et al. [16] conducted experiments using particle image velocimetry (PIV) and discovered that incorporating contoured cavities on the upper surface of the diffuser helps to reduce the extent of the flow separation region, thus enhancing the diffuser's efficiency. They also presented a study that aimed to improve diffuser efficiency using multiple local recirculation regions (MLRR) [17].

The Aerodynamic Shape Optimization Design Group is currently researching optimization methods and developing new techniques and algorithms. In recent studies, they have employed an adjoint-based solver to compute sensitivity derivatives. This approach is demonstrated in two articles: Telidetzki et al. [18] used the B-Spline method to adjust volume parameters, and calculated sensitivity derivatives using an adjoint equation solver and a jet stream solver. This approach was used to obtain an optimized shape for the NACA 0012 airfoil [18].

In a study by Amoignon et al. [19], three methods for the shape optimization of airfoils were compared [19]. The first two methods used free-form deformation, and the third

method used the radial basis function (RBF) coupled with the adjoint method. Sensitivity derivatives were obtained using an adjoint-based solver with an unstable flow solver and a finite difference method. The drag coefficients of the NACA 0012 and RAE 2822 airfoils were reduced under free deformation conditions. In an article by Carrier et al. [20], the Bezier curve technique was used to modify the shapes of the NACA 0012 and RAE 2822 airfoils for shape optimization [20]. The sensitivity derivatives were calculated using a flow solver and an adjoint solver. The conjugate gradient method was used as an optimization algorithm in this study. Carrier et al. [20] achieved a considerable reduction in the pressure drop ratios of NACA 0012 and RAE 2822 using this method [20].

Adjoint optimization has gained popularity, especially in modern industries. In a study by Othmer et al. [21], the method was applied to analyze a transient flow through 2D and 3D ducts using a solver that combined a finite-volume method with continuous-adjoint optimization. Another project was carried out in collaboration with Volkswagen Automotive [22], where the group utilized adjoint equations to improve the aerodynamic shape of various components such as cockpit air ducts, engine components, propulsion, exhaust flow, and the car body. One particular area of interest was the flow around the side mirrors, which involved modifying the mirror surfaces. In 2004, the adjoint method was also used to investigate a multipurpose aerodynamic shape optimization method with a slotted flap [23]. The study utilized a single and multipurpose algorithm and the Newton–Krilov gradient-based method, with twelve moving points on the surface of an airfoil and eight points on the surface of the flap.

In a study by Schramm et al. [24], an adjoint optimization algorithm was employed using the finite difference method to optimize the shape of the front edge (leading edge flaps) of a slotted airfoil, resulting in improved aerodynamic coefficients. Furthermore, the optimized shape was experimentally validated in a wind tunnel. In a separate study, Rashad and Zingg [25] utilized the discrete adjoint method to optimize the aerodynamic shape of an airfoil by simulating natural laminar flow, which included a transient turbulent current under the RANS method. The adjoint method has become a prevalent tool for shape optimization; Elham and van Tooren [26] used this technique in OpenFoam software to optimize the shape of an airfoil [26].

## 3. Methodology

This section presents the proposed algorithm, which consists of several components. First, the NACA four-digit airfoil is introduced. Then, the basic concepts of the adjoint equation are explained, taking into account the flow field, the boundary geometry, and the cost function. The algorithm developed can handle both viscous and non-viscous flows, and in this section, the adjoint equation for viscous flow is described. Finally, the optimization algorithm is presented.

### 3.1. Geometry Generation of the NACA Four-Digit Airfoil

The NACA airfoil was initially developed for aircraft, but it has also found applications in wind turbines. In the literature, the four- and five-digit types of NACA (National Advisory Committee for Aeronautics) have been extensively investigated. The coordinates of the airfoil $(x, y)$ in a plane $(xy)$ are not dimensionless with respect to the chord. Airfoils with NACA four-digit designations have piecewise parabolic mean camber lines $y_c$ and thickness distributions $y_t$ that are consistent across designs. The thickness distribution is given by

$$y_t = \pm 5tc \left( 0.2969\sqrt{x} - 0.1260x - 0.3516x^2 + 0.2843x^3 - 0.1015x^4 \right) \tag{1}$$

where $t$ is the thickness ratio and $c$ is the chord length of the airfoil. The mean camber line $y_c$ comprises two sections, which are defined by different parabolic curves, as in

$$y_c = \begin{cases} \frac{m}{p^2}(2px - x^2) & 0 \le x < p \\ \frac{m}{(1-p)^2}((1-2p) + 2px - x^2) & p \le x \le 1, \end{cases} \tag{2}$$

where $p$ is the maximum camber position along the chord length and $m$ is the camber ratio. In other words, the complete set of the coordinates $(x, y)$ of the NACA airfoil can be described by two curves, i.e., the upper and lower surfaces, which are defined by

$$\begin{aligned} y_{\text{upper}} &= y_c + y_t \cos(\theta) \\ x_{\text{upper}} &= x - y_t \sin(\theta) \\ y_{\text{lower}} &= y_c - y_t \cos(\theta) \\ x_{\text{lower}} &= x + y_t \sin(\theta), \end{aligned} \tag{3}$$

where,

$$\theta = tan^{-1} \frac{dy_c}{dx}. \tag{4}$$

To summarize, the first digit of a NACA airfoil indicates the maximum mean camber, the second digit indicates the position of the maximum camber, and the last two digits indicate the maximum thickness. The NACA4418 airfoil selected as an initial condition for the optimization process described in the following is an asymmetric airfoil with a maximum mean camber of 4% located at a distance of $0.4c$ and a maximum thickness of 18%, which is $t = 0.18$.

### 3.2. Adjoint Equation

The progress made in design procedures relies on the cost function ($I$). Parameters such as the pressure distribution on an object or the lift-to-drag ratio in reverse problems may be used to evaluate the cost function. When dealing with the flow around aerodynamic structures or wings, the cost function ($I$) represents aerodynamic features and involves the flow field $w$ and the boundary geometry $\mathcal{F}$. Therefore, the cost function may be viewed as a function of these two variables.

$$I = I(w, \mathcal{F}). \tag{5}$$

As a result, modifying the boundary $\mathcal{F}$ can cause a corresponding alteration in the cost function

$$\delta I = \left[\frac{\partial I}{\partial w}\right]^T \delta w + \left[\frac{\partial I}{\partial \mathcal{F}}\right]^T \delta \mathcal{F}. \tag{6}$$

The initial term pertains to alterations in the cost function with the flow variables, whereas the subsequent term pertains to modifications in boundary geometry variables. When using control theory, flow field equations should not generate multiple solutions, ensuring the removal of $\delta w$ from Equation (6). If we denote the flow equations as $R$, their general forms can be categorized into two types, based on the flow and geometric variables

$$R = R(w, \mathcal{F}). \tag{7}$$

As a result, modifying the boundary $\mathcal{F}$ leads to

$$\delta R = \left[\frac{\partial R}{\partial w}\right] \delta w + \left[\frac{\partial R}{\partial \mathcal{F}}\right] \delta \mathcal{F} = 0. \tag{8}$$

When applying the finite difference method, one can obtain $\delta w$ from Equation (8) and insert it into Equation (6). However, this approach is not practical in complex and multivariate problems, such as fluid flow fields, and can lead to issues with convergence and obtaining the correct solution. The adjoint method offers an alternative solution by removing $\delta w$-dependent terms from Equation (6). This can be achieved by defining the Lagrangian coefficients $\psi$ and assuming that $\delta R = 0$. Then, $\delta I$ can be expressed as

$$\delta I = \delta I - \psi^T \delta R$$

$$\delta I = \left[\frac{\partial I}{\partial w}\right]^T \delta w + \left[\frac{\partial I}{\partial \mathcal{F}}\right]^T \delta \mathcal{F}$$

$$- \psi^T \left( \left[\frac{\partial R}{\partial w}\right] \delta w + \left[\frac{\partial R}{\partial \mathcal{F}}\right] \delta \mathcal{F} \right) \tag{9}$$

The above sentences can be rearranged as

$$\delta I = \left( \left[\frac{\partial I}{\partial w}\right]^T - \psi^T \left[\frac{\partial R}{\partial w}\right] \right) \delta w$$

$$+ \left( \left[\frac{\partial I}{\partial \mathcal{F}}\right]^T - \psi^T \left[\frac{\partial R}{\partial \mathcal{F}}\right] \right) \delta \mathcal{F}. \tag{10}$$

The arbitrariness of the Lagrangian coefficients $\psi$ allows for the first component of Equation (10) to be zeroed, as

$$\left[\frac{\partial I}{\partial w}\right]^T - \psi^T \left[\frac{\partial R}{\partial w}\right] = 0. \tag{11}$$

By eliminating the dependencies of the changes of the cost function $\delta I$ on the flow variable $\delta w$, only the geometric boundary variable remains. Equation (11) is referred to as the adjoint equation, where the Lagrangian coefficients $\psi$ are the unknowns. When this equation is applied to Equation (10), $\mathcal{G}$, the gradient of the cost function (with respect to the boundary geometric variables) is obtained as follows

$$\mathcal{G} = \frac{\delta I}{\delta \mathcal{F}} = \left[\frac{\partial I}{\partial \mathcal{F}}\right]^T - \psi^T \left[\frac{\partial R}{\partial \mathcal{F}}\right]. \tag{12}$$

*3.3. Adjoint Equation for Viscous Flow*

The equation that describes the overall flow in a domain $\Omega$ can be stated as

$$\frac{\partial w}{\partial t} + \frac{\partial f_i}{\partial x_i} = \frac{\partial f_{vi}}{\partial x_i} \text{ in } \Omega. \tag{13}$$

The vector $w$ corresponds to the state variables, and $f_i$ and $f_{vi}$ are the viscous flux and non-viscous flows for a given dimension $x_i$, respectively. In the case of a two-dimensional flow field that is defined by $x_1 = x$, $x_2 = y$, $f_1 = f$, and $f_2 = g$, the resulting expression is obtained as

$$\frac{\partial w}{\partial t} + \frac{\partial f}{\partial x} + \frac{\partial g}{\partial y} = \frac{\partial f_v}{\partial x} + \frac{\partial g_v}{\partial y} \text{ in } \Omega, \tag{14}$$

where the vectors $g$ and $g_v$ correspond to the viscous and non-viscous fluxes in the $y$ directions, and they are defined as follows

$$w = \begin{Bmatrix} \rho \\ \rho u \\ \rho v \\ \rho E \end{Bmatrix}, \quad f = \begin{Bmatrix} \rho u \\ \rho u^2 + p \\ \rho uv \\ \rho uH \end{Bmatrix}, \quad g = \begin{Bmatrix} \rho v \\ \rho vu \\ \rho v^2 + p \\ \rho vH \end{Bmatrix},$$

$$f_v = \begin{Bmatrix} 0 \\ \sigma_{xx} \\ \sigma_{yx} \\ u\sigma_{xx} + v\sigma_{xy} + k\frac{\partial T}{\partial x} \end{Bmatrix}, \quad g_v = \begin{Bmatrix} 0 \\ \sigma_{xy} \\ \sigma_{yy} \\ u\sigma_{xy} + v\sigma_{yy} + k\frac{\partial T}{\partial y} \end{Bmatrix}. \tag{15}$$

The steady-state non-viscous flow equation by neglecting the viscous sentences of Equation (14) can be written as

$$R = \frac{\partial f}{\partial x} + \frac{\partial g}{\partial y} = 0 \quad \text{in } \Omega. \tag{16}$$

The relationship can be defined as $R$ and can be substituted for the adjoint Equation (11), resulting in

$$
\begin{aligned}
&\left[\frac{\partial I}{\partial w}\right]^T - \psi^T \left[\frac{\partial R}{\partial w}\right] = 0 \quad \text{in} \quad \Omega, \\
&\Rightarrow \left[\frac{\partial I}{\partial w}\right]^T - \psi^T \left[\frac{\partial}{\partial w}\left(\frac{\partial f}{\partial x} + \frac{\partial g}{\partial y}\right)\right] = 0 \quad \text{in} \quad \Omega, \\
&\Rightarrow \left[\frac{\partial I}{\partial w}\right]^T - \psi^T \left[\frac{\partial}{\partial x}\left(\frac{\partial f}{\partial w}\right) + \frac{\partial}{\partial y}\left(\frac{\partial g}{\partial w}\right)\right] = 0 \quad \text{in} \quad \Omega.
\end{aligned}
\tag{17}
$$

Integrating the volume will yield

$$\int_\Omega \left[\frac{\partial I}{\partial w}\right]^T - \int_\Omega \psi^T \left[\frac{\partial}{\partial x}\left(\frac{\partial f}{\partial w}\right) + \frac{\partial}{\partial y}\left(\frac{\partial g}{\partial w}\right)\right] d\Omega = 0 \quad \text{in} \quad \Omega. \tag{18}$$

The second integral, using the partial integrals, is as follows

$$
\begin{aligned}
&\int_\Omega \psi^T \left[\frac{\partial}{\partial x}\left(\frac{\partial f}{\partial w}\right) + \frac{\partial}{\partial y}\left(\frac{\partial g}{\partial w}\right)\right] d\Omega = \\
&\int_B \psi^T \left(\frac{\partial f}{\partial w} n_x + \frac{\partial g}{\partial w} n_y\right) dB - \int_\Omega \left(\frac{\partial \psi^T}{\partial x}\frac{\partial f}{\partial w} + \frac{\partial \psi^T}{\partial y}\frac{\partial g}{\partial w}\right) d\Omega.
\end{aligned}
\tag{19}
$$

The first volume integral of Equation (19) will also be commutable in the form of a cost function. For example, it assumes that the cost function for achieving the optimal pressure on the solid boundary (inverse problem) is defined as

$$I = \frac{1}{2}\int_B (p_{dis} - p_{dis*})^2 dB. \tag{20}$$

In this case, since this cost function is defined on boundary B, its volume integral will be zero, as

$$\int_\Omega \left[\frac{\partial I}{\partial w}\right]^T = 0 \quad \text{in} \quad \Omega. \tag{21}$$

After substituting Equations (21) and (19) into the integral adjoint Equation (18), the resulting expression is obtained as

$$\int_B \psi^T \left(\frac{\partial f}{\partial w} n_x + \frac{\partial g}{\partial w} n_y\right) dB - \int_\Omega \left(\frac{\partial \psi^T}{\partial x}\frac{\partial f}{\partial w} + \frac{\partial \psi^T}{\partial y}\frac{\partial g}{\partial w}\right) d\Omega = 0. \tag{22}$$

The condition for this equation to be zero is that the integers of each integral are zero, separately, as

$$\frac{\partial \psi^T}{\partial x}\frac{\partial f}{\partial w} + \frac{\partial \psi^T}{\partial y}\frac{\partial g}{\partial w} = 0 \quad \text{in} \quad \Omega, \tag{23}$$

$$\psi^T \left(\frac{\partial f}{\partial w} n_x + \frac{\partial g}{\partial w} n_y\right) = 0 \quad \text{in} \quad B. \tag{24}$$

The adjoint Equation (23) is formulated for the entire field with boundary conditions that satisfy Equation (24). To solve the adjoint Equation (23), an artificial term can be added to transform the equation into a time-dependent equation, as

$$\frac{\partial \psi^T}{\partial t} + \frac{\partial \psi^T}{\partial x}\frac{\partial f}{\partial w} + \frac{\partial \psi^T}{\partial y}\frac{\partial g}{\partial w} = 0 \quad \text{in} \quad \Omega. \tag{25}$$

### 3.4. Optimization Algorithm

Based on the discussion of the adjoint equation formation for a fluid problem, the optimization algorithm can be conceptualized in the following manner:

- Step 1: The initial step involves defining the initial parameters for the algorithm, which initiates the primary optimization loop.
- Step 2: The flow field equations are solved to obtain a reliable solution. AUSM$^+$ [27] was employed to compute the flow field variables that will be utilized in subsequent steps.
- Step 3: The adjoint equation is formulated and solved by the algorithm. The algorithm reconstructs the adjoint equation using the flow field variables.
- Step 4: The boundary geometry is deformed in the direction of the maximum steepest descent, adhering to the NACA standard. After observing the trend of movement of the boundary geometry, the shape of the airfoil is altered to approach the optimal condition.
- The field mesh is reproduced, and the algorithm returns to Step 1 to attain the minimum-cost function. In each iteration, the shape undergoes modifications, and a new mesh is produced at the field boundary.

The loop of the algorithm aimed at achieving the optimized shape is depicted in Figure 1. To identify the path of deformation, the relation given by Equation (12) can be employed to enhance the form, as

$$\delta \mathcal{F} = -\lambda \mathcal{G}, \tag{26}$$

where $\lambda$ is the step size. The following equation provides the search direction toward the optimal value using the steepest descent method

$$\mathcal{S} = -\mathcal{G} \tag{27}$$

Determining the step size is also necessary to move in this direction

$$\lambda = \frac{e^{b\ln 10}}{e^{a\ln 10}} = e^{(b-a)\ln 10}, \tag{28}$$

where $a$ and $b$ are the results of

$$a = \begin{cases} 2 & \text{if} \quad \mathcal{S} = 0 \\ \log|\mathcal{S}| & \text{if} \quad \mathcal{S} \neq 0 \end{cases}, \quad b = \begin{cases} -2 & \text{if} \quad \mathcal{F} = 0 \\ \log|\mathcal{F}| & \text{if} \quad \mathcal{F} \neq 0 \end{cases}. \tag{29}$$

To maximize the ratio of the lift-to-drag coefficient, one can combine the objective functions of maximizing the lift coefficient and minimizing the drag coefficient. For this purpose, the coefficients of the general objective function must be calculated as

$$I = \begin{cases} \left(1 - \frac{C_l}{Cl_{\max}}\right)^2 & \text{if} \quad C_d \leq Cd_{\min} \\ \frac{1}{2}\left(1 - \frac{C_l}{Cl_{\max}}\right)^2 + \frac{1}{2}\left(1 - \frac{C_d}{Cd_{\min}}\right)^2 & \text{if} \quad C_d > Cd_{\min} \end{cases}, \tag{30}$$

where $Cd_{\min}$ is the target value of the drag coefficient and $Cl_{\max}$ is the target value of the lift coefficient. As a result, the objective function changes, as

$$\delta I = \begin{cases} -2\left(1 - \frac{C_l}{Cl_{\max}}\right)\frac{\delta C_l}{Cl_{\max}} & \text{if} \quad C_d \leq Cd_{\min} \\ -\left(1 - \frac{C_l}{Cl_{\max}}\right)\frac{\delta C_l}{Cl_{\max}} - \left(1 - \frac{C_d}{Cd_{\min}}\right)\frac{\delta C_d}{Cd_{\min}} & \text{if} \quad C_d > Cd_{\min} \end{cases}. \tag{31}$$

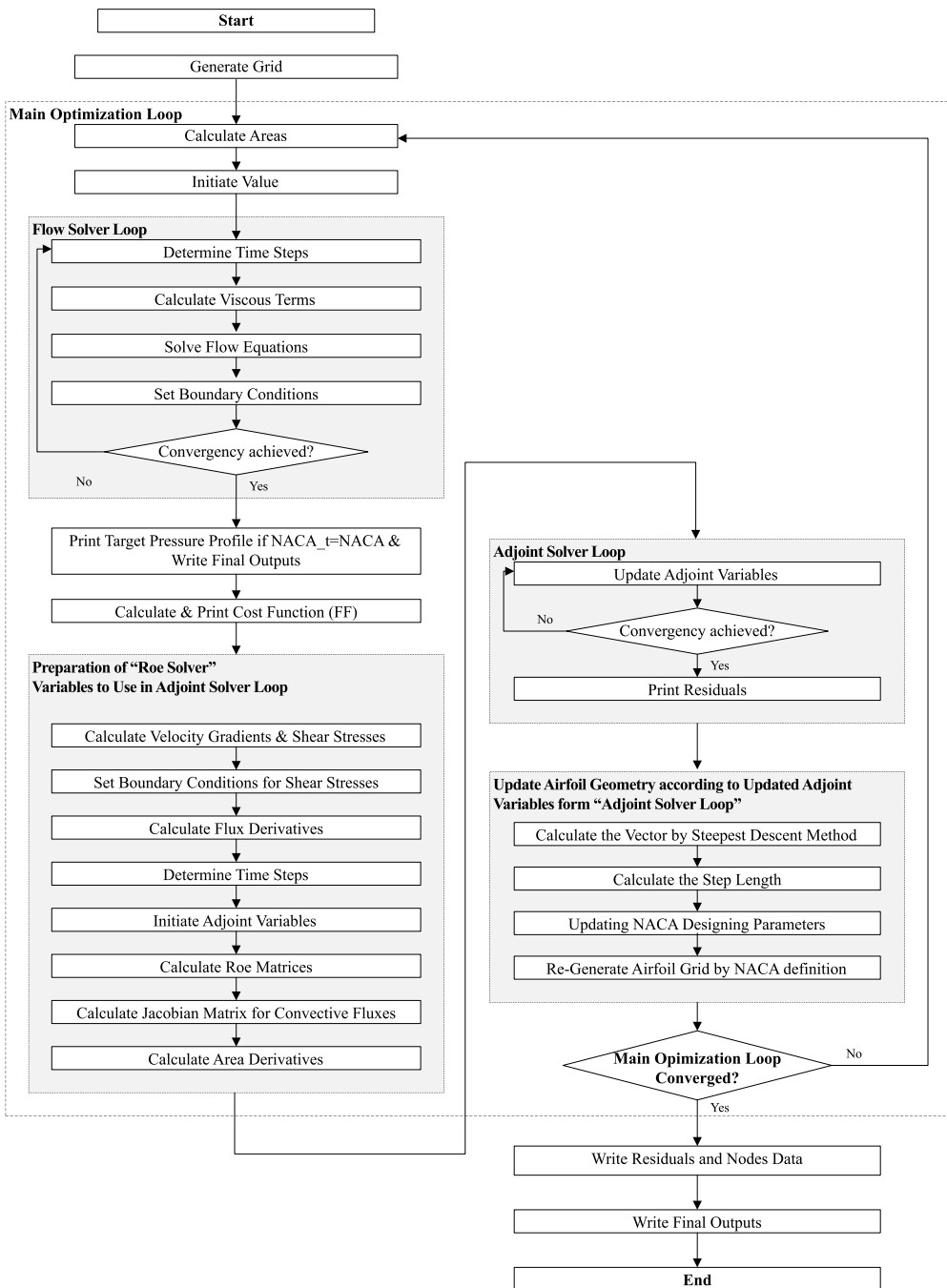

**Figure 1.** The algorithmic flowchart for the shape optimization procedure involves solving the flow field equations, formulating and solving the adjoint equations, computing the converged adjoint variables, and determining the optimization direction based on the adjoint solver variables.

In the optimization process, limitations are set by taking into account the varying abilities of morphing airfoils compared to other airfoils. The range of changes allowed in the determining components of the flap geometry during the optimization path are presented in Table 1.

**Table 1.** The range of changes allowed in the determining components of the flap geometry during the optimization process

| Parameter | Symbol | Ratio |
| --- | --- | --- |
| Airfoil Thickness | $t$ | $0.1\,c \sim 0.4\,c$ |
| Airfoil Maximum Camber | $m$ | 0%~9.5% |
| Airfoil Maximum Camber Position | $p$ | 0.1~0.9 |

## 4. Results

The study case was chosen to assess the accuracy of the proposed algorithm in transient and high-speed flows. The optimization was performed under viscous flow conditions. The adjoint sensibility for 10 different grid generations in one optimization loop, the parameter change rate limitation, the grid convergence analysis for three grid generations by the Roache method, and the results of the multipurpose optimization algorithm are presented.

### 4.1. Study Case Selection

In a previous study, Tanabi et al. [3] evaluated an algorithm under laminar flow conditions [3]. The authors of the current study selected a different sample to evaluate the proposed algorithm under non-viscous conditions. They investigated the performance in transient flow by simulating the flow around a NACA 0012 airfoil with free-flow conditions of $M = 0.85$ and $\alpha = 1°$. These parameters were chosen to assess the accuracy of the proposed algorithm in transient and high-speed flows. The simulation used a circular grid with dimensions of $100 \times 70$.

A circular grid with dimensions of $100 \times 70$ was used to simulate the flow around the airfoil. Figure 2a,b show the lines of the pressure coefficient and Mach number, respectively. The simulation accurately predicted the two shock waves on the surface and below the surface, with an even lower surface shock wave captured, possibly due to the finer size of the grid compared to that used by Liou and Steffen Jr. [28]. The study compared the results of two solver methods, Roe and AUSM, for the pressure coefficient on the airfoil surface shown in Figure 2c, with the numerical results obtained in previous research.

### 4.2. Adjoint Sensitivity and Parameter Changing Range

To validate the adjoint sensitivity, the same input used in the example presented in the paper was applied to different grid conditions. The behavior of the adjoint sensitivity was observed in 10 different runs, which produced similar results as shown in Figure 3. However, additional discussion on this topic may divert the attention from the primary focus of the paper and overwhelm one with additional information. Therefore, we did not address this topic in the paper.

In the optimization process, constraints are established based on the morphing capabilities of the airfoils, which may vary from other types of airfoils. The allowable ranges for modifications to the determining components of the flap geometry in the optimization process are outlined in Table 2.

**Table 2.** The range of changes permitted in the determination of the components of the flap geometry in the optimization path.

| Parameter | Symbol | Ratio |
| --- | --- | --- |
| Airfoil Thickness | $t$ | $0.1\,c$~$0.4\,c$ |
| Airfoil Maximum Camber | $m$ | 0%~9.5% |
| Airfoil Maximum Camber Position | $p$ | 0.1~0.9 |

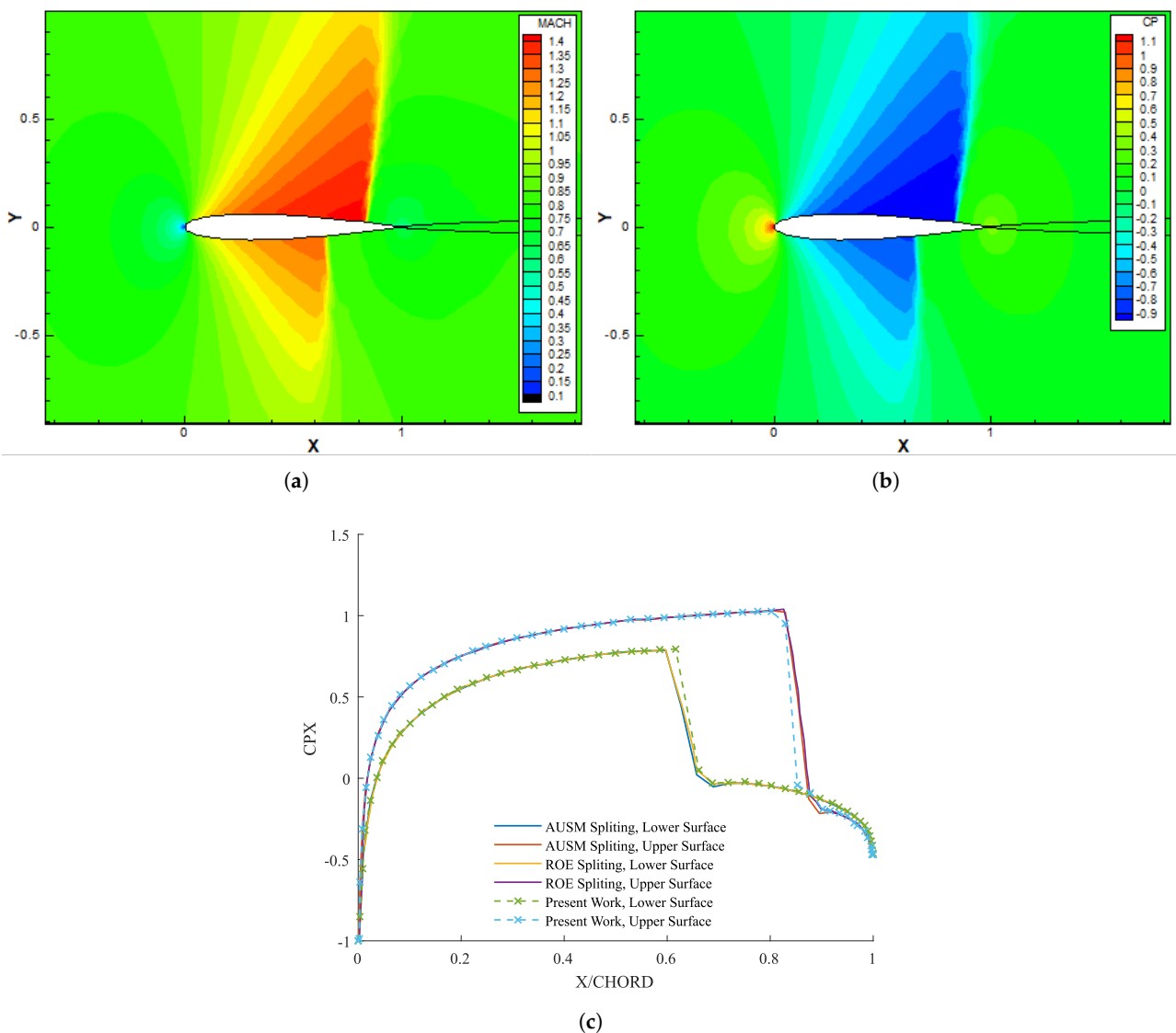

**Figure 2.** (**a**) The Mach lines around the NACA 0012 airfoil under nonlinear flow conditions at $M = 0.8$ and $\alpha = 1°$ are presented. (**b**) Nonlinear pressure coefficient lines are shown around the same airfoil under the same flow conditions. (**c**) The pressure distribution on the surface of the NACA 0012 airfoil is presented for non-viscous flow at $M = 0.8$ and $\alpha = 1°$, and the results are compared to the numerical results obtained by Liou and Steffen Jr. [28].

### 4.3. Grid Convergence Analysis

The order of convergence can be determined using the method proposed by Roache [29], based on the results obtained from testing the developed algorithm with different mesh generations. Figure 4 illustrates the residual of the adjoint variable for the main loop with 2400 iterations using 3 different mesh numbers of $50 \times 35$, $100 \times 70$, and $200 \times 140$. The method provides the means to calculate the order of convergence using

$$p_{GCI} = \ln\left(\frac{f_3 - f_2}{f_2 - f_1}\right) / \ln(r),\tag{32}$$

where $r$ is the constant grid refinement ratio, and $f_1$, $f_2$, and $f_3$ are the lowest residuals of the adjoint variables from cases 1, 2, and 3, respectively. For these test cases, the grid refinement ratio was $r = 2$ and the resulting lowest logarithmic residuals were $\log(f_1) = -3.931$, $\log(f_2) = -3.810$, and $\log(f_3) = -3.796$. The order of convergence of $p_{GCI} = 2.848$ was

obtained for the analyzed data. The method provides the grid convergence index for a given relative difference between two cases using the following expression

$$GCI = \frac{F_S \cdot |\varepsilon|}{r^{p_{GCI}} - 1} \times 100 \tag{33}$$

where $\varepsilon$ is the relative error between two test cases as in $\varepsilon_{i,j} = (f_j - f_i) / f_i$ and $F_S$ is the safety factor. Since three grids were used to estimate $p_{GCI}$, a safety factor of $F_S = 1.25$ was used. The resulting CGI between Case 1 and 2 was $GCI_{1,2} = 0.6566$ and between Case 2 and 3 was $GCI_{2,3} = 4.8875$.

To confirm the accuracy of the solution, the following relationship can be used with the three grids

$$\frac{GCI_{3,2}}{r^p \times GCI_{2,1}} \approx 1. \tag{34}$$

In our case, it is $1.0337 \approx 1$.

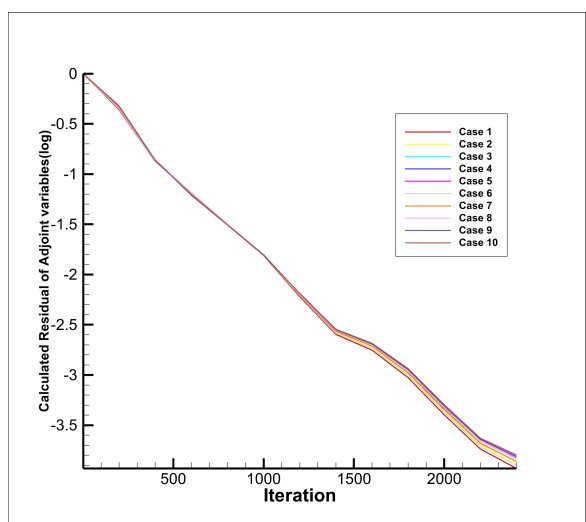

**Figure 3.** The residual of adjoint variables was calculated for 10 runs under the same mesh conditions, using the same input data as the example in the paper. These calculations were performed within a single loop consisting of 2400 iterations.

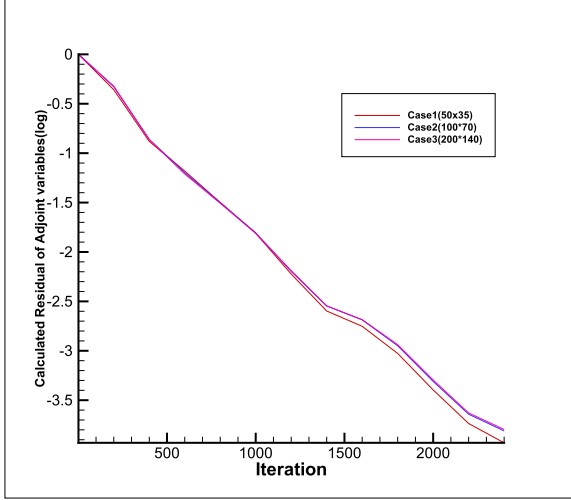

**Figure 4.** The residual of adjoint variables was calculated for three different mesh conditions using the same input data as the example presented in the Multipurpose Optimization section. A single loop consisting of 2400 iterations was used for the calculation.

### 4.4. Multipurpose Optimization of Airfoil

The optimization process involves selecting the NACA4418 airfoil as the primary airfoil and using it to determine the final shape of the airfoil by minimizing the total drag coefficient and maximizing the lift coefficient based on design parameters and the primary airfoil's shape. Optimization is performed under viscous flow conditions as specified in Table 3. The profile design variables include the maximum thickness, maximum camber, and maximum camber position, and their values are adjusted to achieve the intended purpose and obtain the final shape of the profile. The lift and drag values are deliberately chosen in such a way that they cannot be achieved under the defined flight conditions. The purpose of this work is to show the ability of the algorithm in defining the design limits of the shape of the airfoil. which moves toward optimizing the shape as much as possible; after reaching the defined design limits, it continues to optimize so that it can display the most ideal and closest possible shape to the output, according to the optimization goals.

**Table 3.** The flow conditions utilized in the proposed multipurpose shape optimization for the NACA4418 airfoil.

| Parameter | Symbol | Ratio |
|---|---|---|
| Mach Number | M | 0.72 |
| Reynolds Number | Re | 3000 |
| Angle of Attack | $\alpha$ | 2.8° |
| Target Lift Coefficient | $Cl_{max}$ | 0.45 |
| Target Drag Coefficient | $Cd_{min}$ | 0.12 |

Table 4 indicates that although the airfoil thickness and maximum camber decreased, and the camber location changed toward the end of the airfoil, only the drag coefficient was optimized. Therefore, it is not possible to design an airfoil that optimizes both conditions to meet them simultaneously under the specified flow conditions. In Figure 5, the convergence of the cost function, lift-to-drag ratio, and aerodynamic coefficients is illustrated. The graphs show that all parameters have converged to an acceptable value after 20 iterations. The aerodynamic coefficients shown in the figures include the viscous drag coefficient $C_{dv}$, the pressure drag coefficient $C_{dp}$, the lift coefficient $C_l$, the total drag coefficient $C_d$, and the lift-to-drag ratio $C_l/C_d$. Although both optimization goals were not achieved, the changes in all design parameters and aerodynamic coefficients have become insignificant.

**Table 4.** The initial and final values of the components that determine the airfoil geometry and aerodynamic coefficients were optimized to achieve maximum lift and minimum drag coefficients.

| Parameter | Symbol | Initial Value | Final Value |
|---|---|---|---|
| Maximum Thickness | $t$ | 0.180 $c$ | 0.143 $c$ |
| Maximum Camber | $m$ | 4.000% | 3.986% |
| Maximum Camber position | $p$ | 0.400 | 0.397 |
| Lift Coefficient | $C_l$ | 0.101 | 0.398 |
| Pressure Drag Coefficient | $C_{dp}$ | 0.150 | 0.083 |
| Viscous Drag Coefficient | $C_{dv}$ | 0.030 | 0.037 |
| Total Drag Coefficient | $C_d$ | 0.179 | 0.120 |
| Lift-to-Drag Ratio | $C_l/C_d$ | 0.564 | 3.328 |

The authors deliberately chose input values for maximum lift and minimum drag coefficients that were unattainable due to design limitations to evaluate the algorithm's performance. The results show that the algorithm increases the lift-to-drag ratio by reaching the maximum lift target value while minimizing drag, but both goals cannot be achieved simultaneously due to design constraints. Figure 6 illustrates that after fifteen cycles, the maximum thickness $t$, the bend location $m$, and the maximum bend $p$ exhibit insignificant changes.

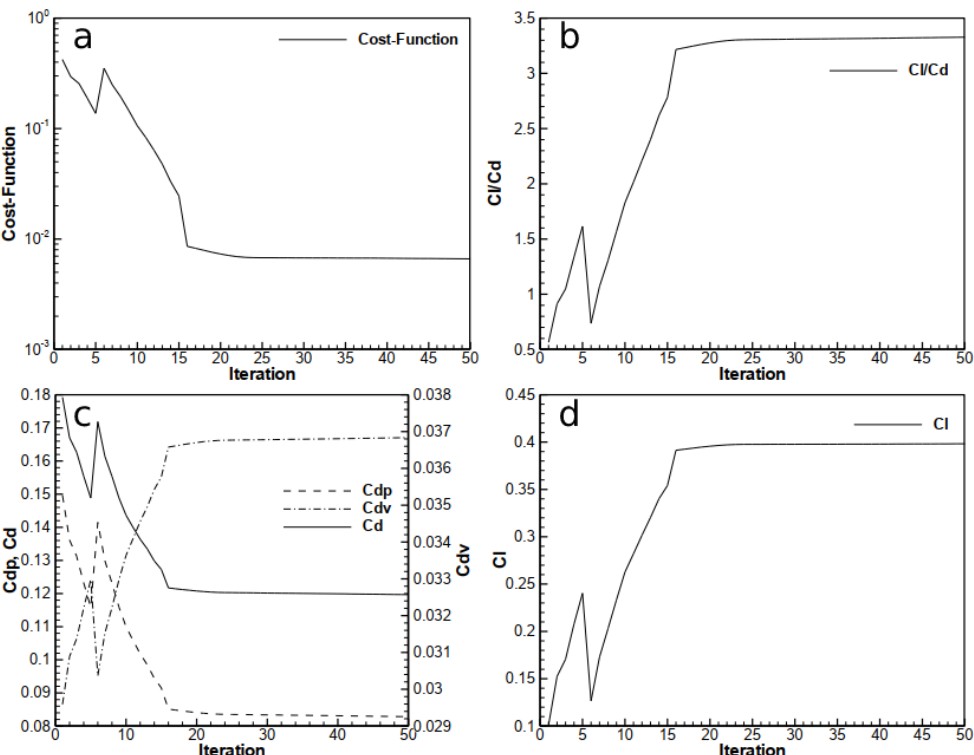

**Figure 5.** The figures illustrate various aspects of the optimization process of the multipurpose lift-to-drag ratio algorithm. (**a**) Changes in the cost function during the optimization process. (**b**) Represents the lift-to-drag ratio $C_l/C_d$ obtained after applying the algorithm. (**c**) Shows variations in the pressure drag coefficient $C_{dp}$, the viscous drag coefficient $C_{dv}$, and the total drag coefficient $C_d$ during the optimization process. Finally, (**d**) displays the changes in the lift coefficient $C_l$ during the optimization process.

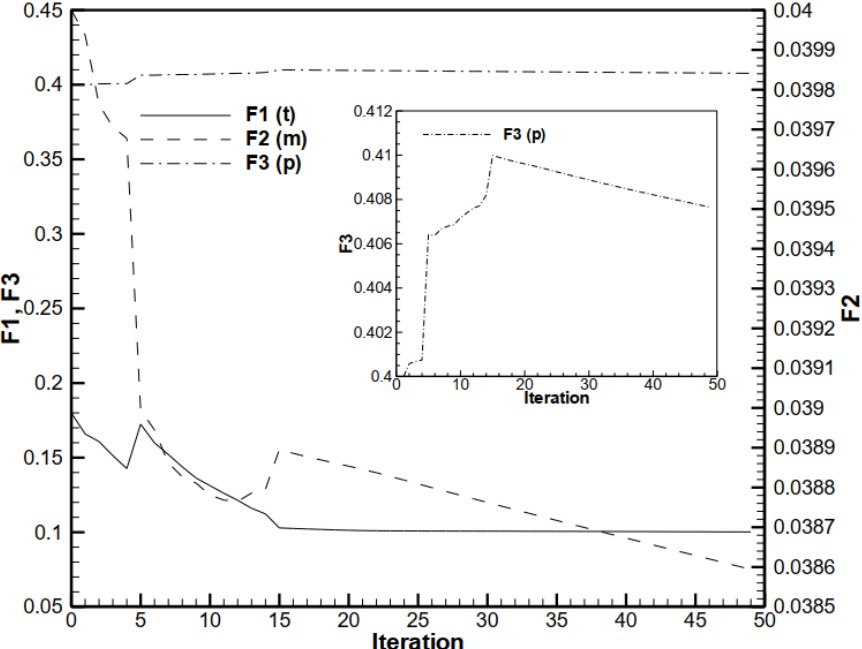

**Figure 6.** The variations in the airfoil design parameters when applying the multipurpose lift-to-ratio optimization algorithm are shown, including the maximum thickness $t$, the location of the bend $m$, and the maximum bend $p$.

The modifications in the pressure coefficient $Cp$ around the airfoil generated by the multipurpose optimization algorithm are shown in Figure 7. The lift coefficient is represented by the enclosed area in this shape. When comparing the distribution of the initial and final airfoil pressure coefficients surrounding the airfoil, it can be seen that the lift coefficient has increased. The shape of the airfoil deformation in the pattern to reach the final shape is illustrated in Figure 8. The algorithm attenuates the thickness until near the design limit (to reach the maximum lift coefficient). After that, it changes other design parameters, such as the maximum camber and position, to achieve the defined goal. The Mach contour around the airfoil is presented in Figure 9. There was a shock in the middle of the airfoil in the initial shape; after optimizing the shape, the shock appeared. The streamlines around the airfoil in the initial and final conditions are shown in Figure 10. The eddies were reduced after optimization, and the separation rate decreased in the airfoil tail, which is evident in the streamlines in the airfoil tail.

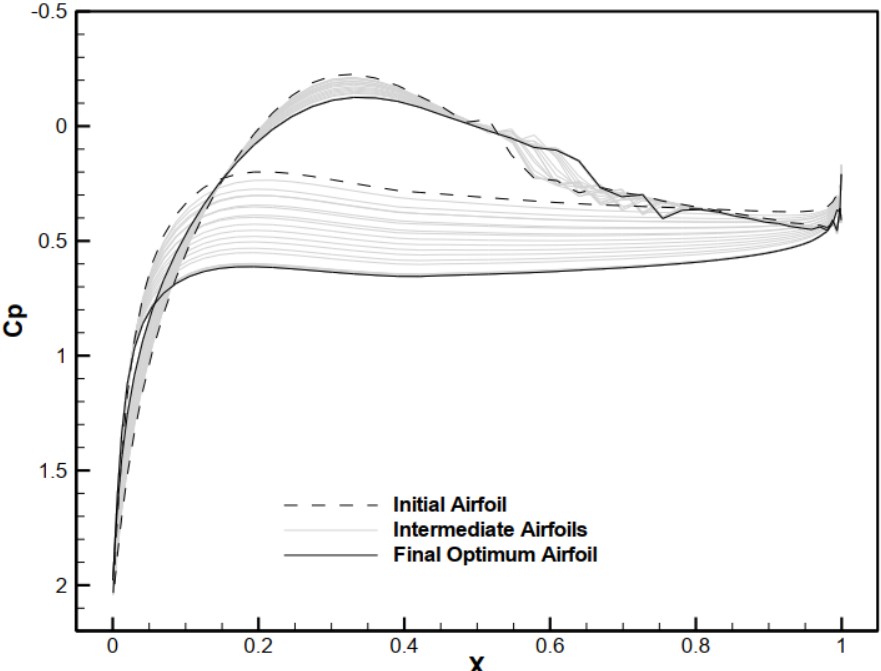

**Figure 7.** The proposed multipurpose optimization algorithm generated modifications in the pressure coefficient $C_p$ around the airfoil.

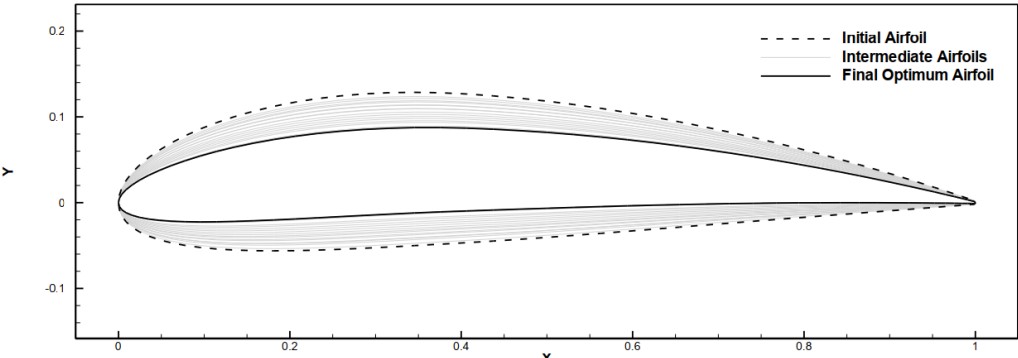

**Figure 8.** The proposed multipurpose optimization algorithm generated modifications in the geometry of the airfoil curve.

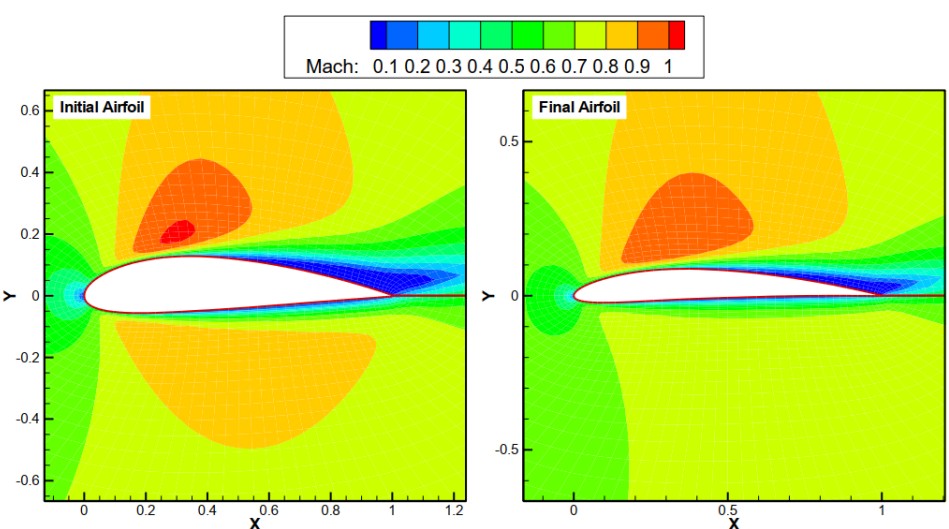

**Figure 9.** The Mach number contour around an airfoil was examined under the initial and final conditions of the proposed multipurpose optimization algorithm.

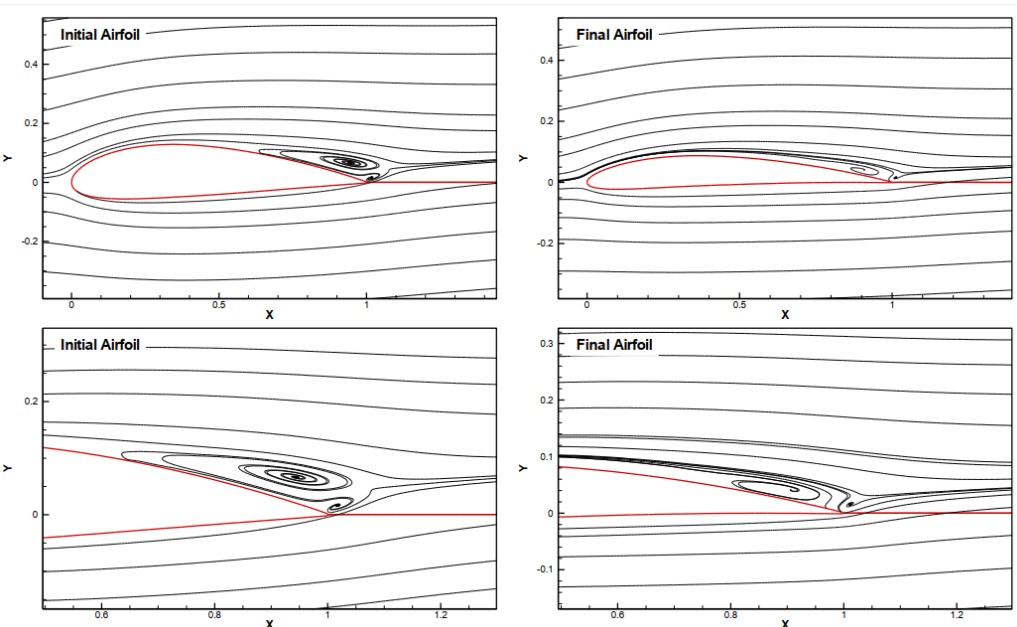

**Figure 10.** The streamlines around the airfoil were observed in both the initial and final conditions of the multipurpose optimization.

## 5. Conclusions

This paper presents an airfoil shape optimization method based on the adjoint approach. The non-viscous flow computations were carried out using Euler equations, while Navier–Stokes equations were used for viscous flow calculations. The AUSM⁺ scheme was utilized to calculate the flow coefficients. The adjoint method was used to apply the optimization equations to the flow equations, resulting in an algorithm for airfoil shape improvement.

As an innovation, multi-objective optimization was performed to evaluate the software performance, in which the lift and drag coefficients were determined as optimization goals. In the issue mentioned above, viscous flow was used. The airfoil design parameters were considered as the maximum thickness, maximum camber, and maximum camber position.

In the multi-objective optimization problem, it was observed that the total drag coefficient achieved the set value; however, the final lift coefficient had a significant difference

from the set value. The outcome suggests that both sets of target values could not be achieved simultaneously under the flow conditions mentioned in the problem. Hence, it is not possible to find every coefficient value under a single flight condition. Consequently, to produce specific lift or drag coefficients, one must modify both the airfoil shape and flow conditions.

**Author Contributions:** Conceptualization, N.T.; methodology, N.T.; software, N.T.; validation, N.T.; formal analysis, N.T.; investigation, N.T.; resources, M.S.G.T.; data curation, N.T.; writing—original draft preparation, N.T. and A.M.S.J.; writing—review and editing, N.T., A.M.S.J. and M.S.G.T.; visualization, N.T.; supervision, M.S.G.T.; project administration, M.S.G.T. and M.A.O.P.; funding acquisition, M.S.G.T. All authors have read and agreed to the published version of the manuscript.

**Funding:** N. Tanabi was supported by Petrobras/ANP/FUSP. M. S. G. Tsuzuki is partially supported by CNPq (Grant 305.959/2016–6). The paper has the support by CAPES/PROAP-Grant 817.757/38.860.

**Institutional Review Board Statement:** Not applicable.

**Informed Consent Statement:** Not applicable.

**Data Availability Statement:** Not applicable.

**Conflicts of Interest:** The authors declare no conflict of interest.

## Nomenclature

| | |
|---|---|
| $\alpha$ | Angle of Attack |
| $\lambda$ | Step Size of Steepest Descent |
| $\mathcal{B}$ | Airfoil's Boundary Geometry |
| $\mathcal{F}$ | Boundary Geometric Variables of Cost Function |
| $\mathcal{G}$ | Gradient of the Cost Function |
| $\mathcal{S}$ | Search Direction of Steepest Descent Method |
| $\Omega$ | Open Domain |
| $\psi$ | Lagrangian Coefficient |
| $\varepsilon$ | Relative Error |
| $a$ | Direction-based Step Size Coefficient |
| $b$ | Geometry-based Step Size Coefficient |
| $c$ | Chord Length |
| $C_{dp}$ | Pressure Drag Coefficient |
| $C_{dv}$ | Viscous Drag Coefficient |
| $C_d$ | Total Drag Coefficient |
| $C_l$ | Lift Coefficient |
| $Cd_{\min}$ | Target Drag Coefficient |
| $Cl_{\max}$ | Target Lift Coefficient |
| $f$ | Viscous Flux |
| $F_S$ | Factor of Safety |
| $f_v$ | Non-viscous Flux |
| $GCI$ | Grid Convergence Index |
| $I$ | Cost Function |
| $M$ | Mach Number |
| $m$ | Maximum Camber |
| $n_x$ | X-component of Normal to B |
| $n_y$ | Y-component of Normal to B |
| $p$ | Maximum Camber Position |
| $p_{dis*}$ | Target Pressure Distribution on Airfoil |
| $p_{dis}$ | Pressure Distribution on Airfoil |
| $p_{GCI}$ | Order of Convergence |
| $R$ | Flow Equations |
| $r$ | Grid Refinement Ratio |

| $Re$ | Reynolds Number |
| --- | --- |
| $t$ | Maximum Thickness |
| $w$ | Flow variables of Cost Function |
| $y_c$ | Mean Camber Line |
| $y_t$ | Thickness distribution |

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
