# Peer review of "Robust Algorithm Software for NACA 4-Digit Airfoil Shape Optimization Using the Adjoint Method"

_applsci, doi:10.3390/app13074269_

Round 1

Reviewer 1 Report

A file with explanation of my conclusions and detailed recommendation is uploaded hereby.

Author Response

The answer to the reviewer is in the attached file.

Reviewer 2 Report

This manuscript is clearly written to show optimization results by an adjoint approach. The method itself is not original but results may be valuable to interested readers. For the completeness, I propose following comments.

- In the "Step 3" in Section 3.4, describe what optimization algorithm is used instead of " in the direction of the maximum steepest descent"

- In Section 4.1, add the validation of adjoint sensitivity.

- In Section 4.2, not clear how multipurpose optimization for lift and drag is performed. How did you combine both parameters?

- In Section 4.2, clearly state the cost function, other constraints and lower/upper bounds of design variables in the standard form of optimization problem.

- Is the final airfoil geometry or a similar one found in other reference airfoil shapes?

Author Response

The answers to the reviewer is in the attached file.

Reviewer 3 Report

The authors study an algorithm to optimize the shape of a NACA 4-digit profile. The aim of the paper is clear and interesting to me. The paper can be considered for publication after the authors have replied to the following mandatory remarks:

1) The authors should provide the grid convergence analysis 

2) The authors should describe the numerical methodology, the turbulence model, the time and space discretization orders, the p-v coupling algorithm, steady/unsteady solver?…

3) A symbol list is strongly recommended.

4) The authors should provide a description of the benefits of their method by comparing it to other approaches, e.g., the optimization codes MOGA-II and the Bezier/splines curves as proposed for internal and external flows e.g. by Poles et al. (2004, 2009), Mariotti et al. (2013, 2015), Brabaint (1984). These procedures and suggested references should also be mentioned among the possible optimization strategies in the introduction of the paper. 

References:

S. Poles et al. (2009) The effect of initial population sampling on the convergence of multi-objective genetic algorithms. Multiobjective Programming and Goal Programming 3, Springer, Berlin, Heidelberg, 123–133.

S. Poles et al. (2004) MOGA-II performance on noisy optimization problems. Proceedings of the International Conference on Bioinspired Optimization Methods and their Applications, Jozef Stefan Institute, Ljubljana, 51–62.

A. Mariotti et al. (2013) Separation control and efficiency improvement in a 2D diffuser by means of contoured cavities. Eur. J. Mech. B/Fluids 41, 138-149.

A. Mariotti et al. (2015) Use of multiple local recirculations to increase the efficiency in diffusers. Eur. J. Mech. B/Fluids 50, 27–37.

V. Braibant, C. Fleury (1984) Shape optimal design using B-splines. Comput. Meth. Appl. Mech. Eng. 44(3), 247-267.

Author Response

(The authors gave the same response as above.)

Round 2

Reviewer 1 Report

A file has been uploaded hereby.

Author Response

The list of symbols was revised. The text was carefully revised as pointed by the reviewer.

Reviewer 3 Report

Accept in present form

Author Response

Thank you very much.